# StrokeRehab: A Benchmark Dataset for Sub-second Action Identification

**Aakash Kaku**[*1], **Kangning Liu**[*1], **Avinash Parnandi**[*2], **Haresh Rengaraj Rajamohan**[1],
**Kannan Venkataramanan**[1], **Anita Venkatesan**[2], **Audre Wirtanen**[2], **Natasha Pandit**[2],
**Heidi Schambra**[†2], **Carlos Fernandez-Granda**[†1,3]
[1]NYU Center for Data Science [2]NYU School of Medicine
[3] Courant Institute of Mathematical Sciences

## Abstract

Automatic action identification from video and kinematic data is an important machine learning problem with applications ranging from robotics to smart health. Most existing works focus on identifying coarse actions such as running, climbing, or cutting vegetables, which have relatively long durations and a complex series of motions. This is an important limitation for applications that require identification of more elemental motions at high temporal resolution. For example, in the rehabilitation of arm impairment after stroke, quantifying the training dose (number of repetitions) requires differentiating motions with sub-second durations. Our goal is to bridge this gap. To this end, we introduce a large-scale, multimodal dataset, StrokeRehab, as a new action-recognition benchmark that includes elemental short-duration actions labeled at a high temporal resolution. StrokeRehab consists of high-quality inertial measurement unit sensor and video data of 51 stroke-impaired patients and 20 healthy subjects performing activities of daily living like feeding, brushing teeth, etc. Because it contains data from both healthy and impaired individuals, StrokeRehab can be used to study the influence of distribution shift in action-recognition tasks. When evaluated on StrokeRehab, current state-of-the-art models for action segmentation produce noisy predictions, which reduces their accuracy in identifying the corresponding sequence of actions. To address this, we propose a novel approach for high-resolution action identification, inspired by speech-recognition techniques, which is based on a sequence-to-sequence model that directly predicts the sequence of actions. This approach outperforms current state-of-the-art methods on StrokeRehab, as well as on the standard benchmark datasets 50Salads, Breakfast, and Jigsaws.

## 1   Introduction

In domains ranging from robotics to smart health, automatically identifying action from video and kinematic data is an important machine learning problem. In some of these applications it is critical to identify motions at high temporal resolution. This is the case in data-driven stroke rehabilitation, which requires classifying and counting sub-second single-goal motions. In order to advance methodology for high-resolution action identification, it is crucial to establish appropriate benchmark datasets. Existing benchmarks, such as 50Salads [52], Breakfast [32], Jigsaws [18], or Kinetics [28] contain very few short-duration actions (see Figure 3). To address this, we introduce a large-scale, multimodal dataset, StrokeRehab, as a new action-recognition benchmark that includes elemental short-duration actions labeled at a high temporal resolution. The dataset consists of high-quality wearable sensor and video data of 51 stroke patients and 20 healthy subjects. These

---

[*]Equal Contribution
[†]Joint Last Author

36th Conference on Neural Information Processing Systems (NeurIPS 2022) Track on Datasets and Benchmarks.

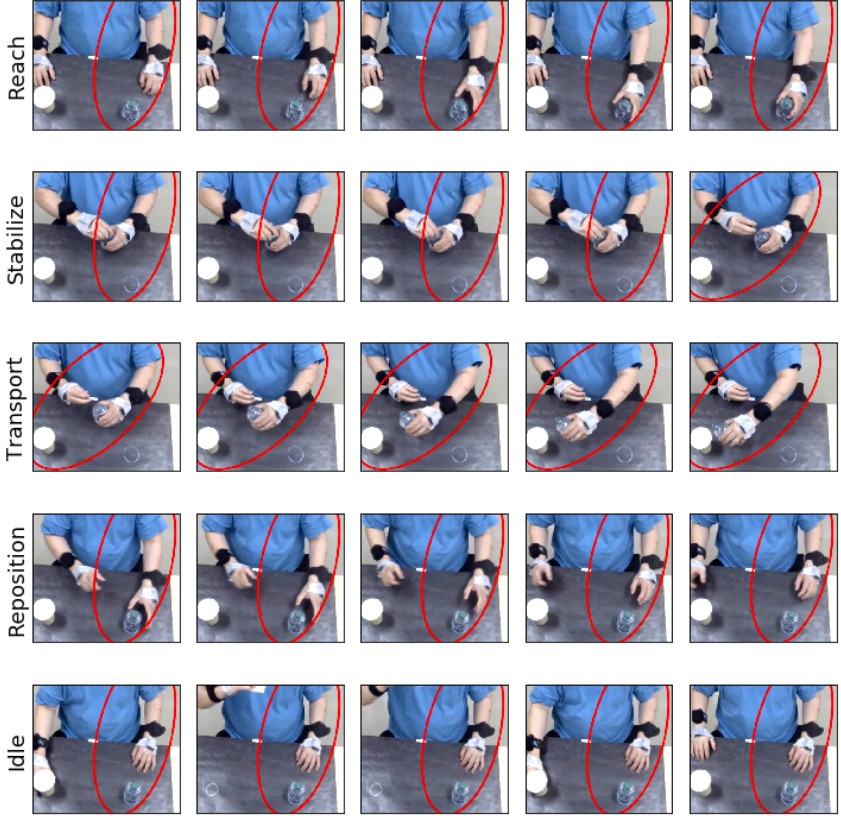

Figure 1: A stroke patient performing a functional activity (drinking) from the StrokeRehab activities battery. Using the functional motion taxonomy, the activity can be decomposed into its constituent functional primitives as follows: *reach*, upper extremity (UE) motion to bring it into contact with a target object (e.g. water bottle); *stabilize* minimal UE motion to keep a target object still (e.g. holding the bottle to allow the other UE to open the cap); *transport*, UE motion to move a target object (e.g. moving the bottle to pour some water); *reposition*, UE motion proximate to a target object (e.g. to move the to the initial neutral spot); *idle*, minimal UE motion to stand at the ready near a target object.

subjects performed nine activities of daily living like drinking, eating, applying deodorant, etc. in a rehabilitation gym (see Figure 1 for an example). The elemental actions performed by the subjects in each session were meticulously labeled by trained annotators overseen by an expert, who examined one third of their labels. These labeled actions are called functional primitives, consisting of five main classes: reach (upper extremity (UE) motion to make contact with a target object), reposition (UE motion to move into proximity of a target object), transport (UE motion to convey a target object in space), stabilization (minimal UE motion to hold a target object still), and idle (minimal UE motion to stand at the ready near a target object) [48].

Evaluation of current state-of-the-art models for action segmentation on StrokeRehab reveals that these approaches are not as effective when applied to short-duration elemental actions. The reason is that the boundaries of these actions are not clearly defined, even for human-expert annotators. As a result, segmentation-based approaches produce noisy estimates, which limits their accuracy. To address this limitation, we introduce an approach to action identification inspired by speech-recognition models, which achieves state-of-the-art performance on StrokeRehab, and also outperforms existing approaches on the standard benchmark datasets like Breakfast, 50Salads, and Jigsaws for the task of sequence-identification. This showcases how StrokeRehab can contribute to methodological advances in machine-learning methodology for action identification.

StrokeRehab contains data from healthy and stroke-impaired individuals. It therefore provides a real-world example of distributional shift. We show that models trained on impaired patients are able to generalize to healthy subjects, but the opposite is not true. In addition, we show that the

performance of models trained on moderately-impaired patients does not generalize effectively to severely-impaired patients. StrokeRehab therefore provides a challenging benchmark dataset to evaluate methods addressing distributional shift.

StrokeRehab also has the potential to advance data-driven stroke rehabilitation. Basic research in animals indicates that the repeated practice of functional motions early after stroke markedly boosts recovery [25, 4, 42]. The same is believed to be true for humans undergoing rehabilitation for stroke-induced disability. However, there has been no systematic quantification of how many training repetitions are needed early after stroke for optimal recovery [31]. Data-driven quantification requires identifying and counting motions at high temporal (sub-second) resolution. StrokeRehab provides a high-quality labeled benchmark dataset for this task.

To summarize, StrokeRehab is a multimodal, labeled real-world dataset, which provides a benchmark for (1) sub-second action identification, (2) generalization in the presence of realistic distributional shift, and (3) data-driven quantification of rehabilitation dose.

## 2 Description of the dataset

### 2.1 Clinical motivation

Stroke is the leading cause of disability in the United States. It affects nearly 800,000 individuals per year, with the numbers of stroke cases increasing as our population ages [19, 46, 6]. Stroke affects the arm in 77% of patients, causing long-lasting motor impairment [36, 34]. By six months, most of these patients remain unable to independently perform activities of daily living, such as feeding, bathing, grooming, etc. This loss of independence reduces the quality of life of both the patients and their caretakers [45, 7] and exacts a heavy societal toll, with annual caretaking and healthcare costs predicted to skyrocket to $240 billion by 2030 [23]. Due to the profound impact of stroke on arm function and its downstream consequences, we focus on the arm in our study.

Following stroke, some spontaneous recovery occurs because of brain plasticity, but this plasticity alone does not fully restore function. In animal models of stroke, training high numbers of functional arm motions not only increases this plasticity [29, 4], but also markedly boosts recovery [42, 25]. It is increasingly believed that if intensive rehabilitation training can be delivered early after stroke in humans, recovery could be similarly accelerated [31].

In rehabilitation training, patients use their impaired arm to practice activities of daily living (ADLs). ADLs are composed of five elemental actions called functional primitives: reach, transport, reposition, stabilize, and idle [48]. For example, in a drinking activity, our arm would "idle" as it rests at our side, then "reach" for a glass and "transport" the glass to our mouth, then "transport" the glass back to the table, and finally "reposition" back to our side for an "idle." Examples of these primitives for a drinking activity can be seen in Figure 1.

A major clinical question is how many repetitions of functional motions are needed to boost recovery. In animal research, the number of repetitions that promote recovery has been quantified [25]. For humans, this quantification has not been done. A handful of studies have observed that patients train about 10 times fewer repetitions than what recovering animals receive, suggesting pronounced under-training in human rehabilitation [35, 30]. However, the optimal number of training repetitions to boost recovery remains uncertain in humans. Currently, the best way to quantify rehabilitation is hand tallying. If performed in real time by the rehabilitation therapist, hand tallying distracts from treatment delivery. If the session is videotaped and annotated offline, tallying is laborious and slow: **it takes one hour of manual effort to label one minute of recorded training**. Hand tallying thus incurs time, effort, and personnel costs that render it unscalable.

Therefore, to facilitate the quantification of training motions in stroke rehabilitation, we developed an approach that combines unobtrusive motion capture with automated identification. We collected the StrokeRehab dataset that consists of labeled sensor and video data from stroke patients and healthy subjects. We then used the StrokeRehab dataset to train models to automatically identify and count functional primitives. The dataset is hosted at SimTK website - `https://simtk.org/projects/primseq`. The license agreement and datasheet for the dataset can be found in Appendix I and Appendix J, respectively.

### 2.2 Cohort selection

We collected sensor and video data from 51 stroke patients and 20 healthy subjects in an inpatient rehabilitation gym.

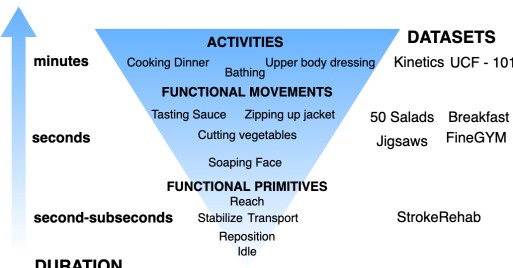

Figure 2: Action recognition datasets ordered according to a hierarchy of the labeled actions they contain. Our dataset StrokeRehab consists of short-duration elemental actions, called functional primitives.

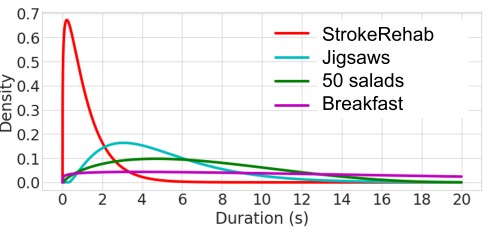

Figure 3: Distribution of action duration for various benchmark datasets and StrokeRehab. This illustrates the extreme fine-grained nature of actions (functional primitives) in the StrokeRehab dataset in comparison to existing ones.

Stroke patients: Individuals were included if they were $\geq$ 18 years old, premorbidly right-handed, and had unilateral arm weakness from an ischemic or hemorrhagic stroke that occurred at least 6 months prior.

Patients were excluded if they had traumatic brain injury; any musculoskeletal or non-stroke neurological condition that interferes with motor function; contracture at the shoulder, elbow, or wrist; moderate arm dysmetria or truncal ataxia; visuospatial neglect; apraxia; global inattention; or legal blindness. A trained assessor quantified arm impairment with the upper extremity Fugl-Meyer Assessment, where a maximum score of 66 signifies no impairment [16]. Stroke impaired patients were divided into two sub-cohorts: moderately to mildly impaired (scores 23-65) and severely impaired patients (scores 8-23). For healthy controls, individuals were included if they were $\geq$ 18 years old, had a right-handed dominance and no motor impairment (score = 66).

Table 1 describes the demographic and clinical characteristics of the stroke impaired patients and healthy controls.

Table 1: Demographic and clinical characteristics of the stroke impaired patients and healthy controls in the cohort. Mean and ranges in parentheses are shown. The cohort is divided into a training set and a test set of mildly and moderately-impaired patients. There is no overlap of patients between the training and test set.

|  | Training set (Mild + Moderate) | Test set (Mild + Moderate) | Severe set | Healthy control |
|---|---|---|---|---|
| n | 35 | 8 | 8 | 20 |
| Age (in years) | 56.56 (21.2-82.7) | 60.8 (42.6-84.2) | 59.73 (41-74.3) | 62.47 (42-82.9) |
| Gender (Female : Male) | 19 F : 16 M | 4 F : 4 M | 5 F : 3 M | 9 F : 11 M |
| Time since stroke (in years) | 6.5 (0.3-38.4) | 3.1 (0.4-5.7) | 3.46 (1.14-6.43) | NA |
| Paretic side (Left : Right) | 20 L : 15 R | 4 L : 4 R | 4 L : 4 R | NA |
| Fugl-Meyer Assessment score | 48.1 (26-65) | 49.4 (27-63) | 16 (8-23) | 66 |

## 2.3 Data acquisition and labelling

Upper body motion was recorded while subjects performed activities of daily living commonly used during stroke rehabilitation. The activities included: washing the face, applying deodorant, combing the hair, donning and doffing glasses, preparing and eating a slice of bread, pouring and drinking a cup of water, brushing teeth, and moving an object horizontal and vertical target arrays. Subjects performed five repetitions of each activity. See Appendix H.1 for detailed descriptions of the activities.

**Description of kinematic data**: Upper body motion was recorded using nine Inertial Measurement Units (IMUs, Noraxon, USA) attached to the upper body, specifically the cervical vertebra C7, the thoracic vertebra T12, the pelvis, and both arms, forearms, and hands (see Figure 8 in Appendix H). The sensors are lightweight (34 g) and small (matchbook-size). They are adhered to the back of the hand with thin tape that does not interfere with finger movement or grasp. Similarly, the straps holding the sensors to the forearms and arms do not cross any joints. Neither the location nor the methods used to affix the sensors are expected to interfere with natural motion. These IMUs captured

76-dimensional kinematic features of 3D linear accelerations, 3D quaternions, and joint angles from the upper body (see Appendix H.2 for details). As an additional feature, for stroke-impaired patients, we included the paretic (stroke-impaired) side of the patient (left or right) encoded in a one-hot vector, increasing the dimension of the feature vector to 77. For healthy subjects, the movements of both hands are labeled. Therefore, the 77th feature equals *right* if we are making predictions for the right hand and *left* if we are making predictions for the left hand.

Each IMU captures 3D linear accelerations and angular velocities at 100 Hz. Angular velocities are converted to sensor-centric unit quaternions, representing the rotation of each sensor on its own axes, with coordinate transformation matrices. In addition, proprietary software (Myomotion, Noraxon) generates 22 anatomical angle values using a rigid-body skeletal model scaled to patient height. See Appendix H.2 for a detailed description of these angles. Each entry (except for the feature encoding paretic side) was mean-centered and normalized separately for each task repetition in order to remove spurious offsets introduced during sensor calibration.

**Description of video data**: Video data were synchronously captured using two high definition cameras (1088 x 704, 60 frames per second or 100 frames per second; Ninox, Noraxon) placed orthogonally < 2 m from the subject. We extract frame-wise features from the raw videos using the X3D model [14], a 3D convolutional network designed for video classification. The model is pretrained on the Kinetic dataset [28], which consists of coarse actions like running, climbing, sitting, etc. Since the StrokeRehab dataset consists of elemental, sub-second actions, we fine-tuned the X3D model on the training set of StrokeRehab. In order to fine-tune, we used video sequences as input and trained the model to identify the primitive happening in the center frame of the sequence.

**Data labeling**: The elemental actions performed by the patients in each session were meticulously labeled by trained annotators overseen by an expert, who examined one-third of their labels. Interrater reliability between the coders and expert was high, with Cohen's kappa $\geq 0.96$ between the coders and the expert.

## 2.4 Training and test sets

**Healthy subjects.** We randomly assigned 16 subjects to a training set and 4 subjects to the *Healthy subject test set*.

**Stroke patients.** We used a sub-cohort of 43 mild and moderately impaired patients to create the stroke patient training and test sets. Patients were separated into eight subgroups, balancing for impairment level and paretic side (left or right). One patient in each group was randomly removed and assigned to the *Stroke patient test set*.

**Severely impaired stroke patients.** A second sub-cohort of 8 severely impaired patients with impairment scores in the range 8-23 [16] was used to create a challenging test set to evaluate model generalizability to patients with higher impairment levels. We call this the *Severe impairment test set*.

## 2.5 Data/Label quality

An expert in the functional motion taxonomy (AP) [48] individually trained the annotators, who underwent one month of intensive training on a series of increasingly complex activities. Once annotators labeled the training videos with high accuracy (less than 2% errors), they were given new videos to label independently. The annotators identified and labeled functional primitives in the video, which simultaneously labeled primitives in the IMU data. To ensure consistent labeling, the expert inspected one-third of all labeled videos. Inter-rater reliability between the annotators and expert was high across primitives (Cohen's kappa: reaches, 0.96; repositions, 0.97; transports, 0.97; stabilizations, 0.98; idles, 0.96). When computed in terms of Action Error Rate (AER in Section 3.2), the inter-rater reliability for the entire dataset is 0.021. One minute of recording took on average 79.8 minutes to annotate. There were two experts and seven annotators.

## 2.6 Comparison to existing benchmark datasets

The dataset consists of 3,372 trials of rehabilitation activities performed by 51 stroke-impaired and 20 healthy subjects. Cumulatively, they performed 120,891 functional primitives, which is more than existing benchmark datasets such as FineGym (32,697 annotated sub-actions), Breakfast (11,656 annotated actions), Jigsaws (1,701 annotated actions), and 50Salads (999 annotated actions). A non-trivial amount of manual effort (approximately 2,700 human hours) was required to label the functional primitives, which span 43.48 hours of recorded training.

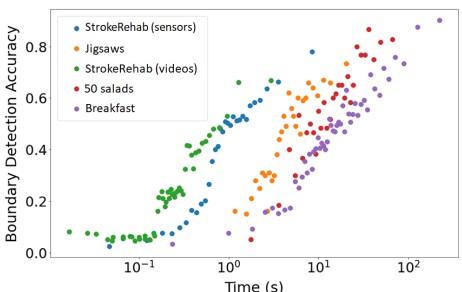

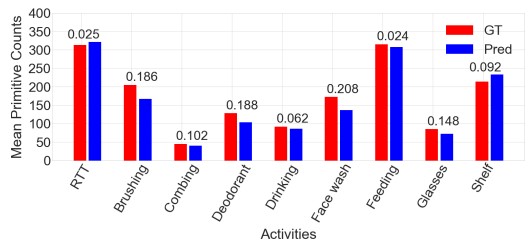

Figure 4: Boundary accuracy achieved by the segmentation models vs duration of the actions for several datasets. Boundary-detection accuracy is directly proportional to action duration.

Figure 5: Comparison of ground-truth and predicted mean counts for the different high-level activities in the StrokeRehab dataset. The relative error is very small for structured activities like moving objects on/off a shelf (Shelf), and larger for unstructured activities like brushing.

Figure 2 shows a hierarchy of human actions. Existing datasets focus on high level actions associated to particular activities or objects (e.g. cutting vegetables, zipping up jacket), which typically have long durations (seconds or longer) and execute several goals. In contrast, as shown in Figure 3, the actions in the StrokeRehab dataset are much shorter (sub-second). They correspond to functional primitives that execute one goal, and are therefore qualitatively different to the higher-level actions in existing benchmark datasets. In particular, they are not associated to specific objects or contexts; identifying them correctly requires learning to distinguish elemental human movements.

## 3 Action sequence identification

The main task associated to StrokeRehab is action sequence identification, i.e. identifying the correct sequence of actions carried out by an individual. This is often the ultimate goal in applications of action recognition [41, 43, 1], and is particularly critical in data-driven rehabilitation [26].

### 3.1 Problem definition

Let $\mathbf{x} = (x_1, ..., x_T)$ be an input sequence with length $T$, which may correspond for example to high-dimensional sensor or video features. The goal of action sequence identification is to estimate the corresponding sequence of actions encoded as $\mathbf{y} = (y_1, ..., y_{T'})$, where $y_i$ $(1 \leq i \leq T')$ is one of $c$ different actions, so $y_i \in \{1, \ldots, c\}$. The length $T'$ of this sequence is much shorter than the input sequence because each action takes place over several time steps. For example, a 100-frame video sequence from the StrokeRehab dataset may correspond to the 3-action sequence: *reach*, *transport*, *stabilize*.

### 3.2 Evaluation metric: Action error rate (AER)

In this section we introduce a metric to evaluate methods for action sequence identification based on the Levenshtein distance, which is a distance tailored to sequence estimation tasks. The Levenshtein distance $\mathrm{L}(G, P)$ between a ground-truth sequence $G$ and a predicted sequence $P$ is the minimum number of insertions, deletions, and substitutions required to convert $P$ to $G$. For example, if $G =$ [*reach*, *idle*, *stabilize*] and $P =$ [*reach*, *transport*], then $\mathrm{L}(G, P) = 2$ (*transport* is substituted for *idle* and *stabilize* is inserted). An important consideration is how to normalize this distance in order to obtain a metric to evaluate sequence identification. Existing works in action recognition [13, 57, 24, 38] use the edit score (ES):

$$\mathrm{ES}(G, P) := \left( 1 - \frac{\mathrm{L}(G, P)}{\max(\mathrm{len}(G), \mathrm{len}(P))} \right) \times 100, \tag{1}$$

where $\mathrm{len}(G)$ and $\mathrm{len}(P)$ are the lengths of $G$ and $P$ respectively. Due to the normalization factor consisting of the maximum between the estimated and ground-truth sequences, this metric is lenient with long estimated sequences containing noisy, spurious estimates. This can be addressed by normalizing with respect to the length of the ground-truth sequence. We call the corresponding metric action error rate (AER), since it is analogous to word error rate, a standard metric in speech

Table 2: Results for action sequence identification on the stroke-patient test set of StrokeRehab (top table) and on existing benchmark datasets (bottom table). We report the mean of the metrics of interest with 95% confidence intervals computed via bootstrapping (see Appendix G.2). In both tables, * indicates models selected based on the best validation frame-wise accuracy. Overall, we observe that seq2seq models tend to outperform segmentation-based approaches. Additional metrics like true positive rate and false discovery rate can be seen in Appendix D

| | | StrokeRehab (stroke-patient test set) | | | |
|---|---|---|---|---|---|
| | Model | Video Data | | Sensor Data | |
| | | Edit Score | Action Error Rate | Edit Score | Action Error Rate |
| Segmentation-based model | MS-TCN* [13] | 60.7 (59.1 - 62.2) | 0.408 (0.388 - 0.428) | 66.9 (65.0 - 68.9) | 0.372 (0.335 - 0.406) |
| | MS-TCN [13] | 62.2 (60.8 - 63.6) | 0.392 (0.371 - 0.413) | 68.7 (67.3 - 70.5) | 0.330 (0.307 - 0.354) |
| | + Smoothing window | 62.7 (61.3 - 64.1) | 0.390 (0.370 - 0.410) | **68.8 (67.3 - 70.3)** | 0.317 (0.297 - 0.338) |
| | ASRF* [24] | 56.9 (55.2 - 58.6) | 0.449 (0.427 - 0.472) | 68.2 (66.7 - 69.9) | 0.328 (0.309 - 0.348) |
| | ASRF [24] | 58.7 (57.3 - 60.2) | 0.436 (0.417 - 0.456) | 67.9 (66.3 - 69.5) | 0.349 (0.326 - 0.372) |
| seq2seq | Seg2seq | **67.6 (66.4 - 68.8)** | **0.322 (0.307 - 0.339)** | 63.0 (61.3 - 64.7) | 0.337 (0.311 - 0.363) |
| | Raw2seq | 66.6 (65.4 - 67.9) | 0.329 (0.312 - 0.345) | **68.8 (67.4 - 70.3)** | **0.305 (0.284 - 0.324)** |

| | | Existing benchmark datasets | | | | | |
|---|---|---|---|---|---|---|---|
| | Model | 50 salads | | Breakfast | | Jigsaws | |
| | | Edit Score | Action Error Rate | Edit Score | Action Error Rate | Edit Score | Action Error Rate |
| Segmentation-based model | MS-TCN* [13] | 68.8 | 0.47 | 62.0 | 1.16 | 55.24 | 0.96 |
| | MS-TCN [13] | 70.8 | 0.43 | 61.7 | 0.97 | 61.44 | 0.82 |
| | + Smoothing window | 76.4 | 0.32 | 69.1 | 0.51 | 76.54 | 0.31 |
| | ASRF* [24] | 74.0 | 0.34 | 71.2 | 0.44 | 71.29 | 0.37 |
| | ASRF [24] | 75.2 | 0.33 | 70.9 | 0.45 | 74.63 | 0.31 |
| Seq2seq | Seg2seq | **76.9** | **0.30** | **73.7** | **0.37** | **83.87** | **0.17** |
| | Raw2seq | 69.4 | 0.54 | 64.1 | 0.55 | 70.13 | 0.35 |

recognition:

$$\text{AER}(G, P) := \frac{\text{L}(G, P)}{\text{len}(G)} \tag{2}$$

AER penalizes longer and shorter predictions equally. For example, if $G$ = [*reach*, *idle*, *stabilize*], $P_1$ = [*reach,idle*], and $P_2$ = [*reach*, *idle*, *stabilize*, *transport*], then ES$(G, P_1)$ = 0.67 and ES$(G, P_2)$ = 0.75, but AER$(G, P_1)$ = AER$(G, P_2)$ = 0.33.

### 3.3 Limitations of segmentation-based models

As described in Appendix A, where we provide a detailed description of the state of the art, most existing methods [13, 57, 24, 38] address the task of action sequence identification by performing segmentation of the input data and then removing label repetitions at consecutive steps (see Figure 6 for a concrete example). Experimentally we observe (see Section 3.4) that this may result in systematically over-segmented noisy outputs, as reported also in [13, 57, 24]. Recent works [57, 24] address this by training a separate action-boundary detection network. The boundary predictions are then used to refine the frame-wise predictions. As pointed out by [49], boundaries of high-level actions are more detectable because the adjacent motions are distinctive; for example, the motions associated with cutting tomatoes versus tossing a salad are very different. In contrast, boundaries of fine-grained actions are harder to identify because their transitions are more elemental; for example, the boundary between the end of a *reach* and the beginning of a *transport* in the StrokeRehab dataset is determined by when the finger pads have fully contacted a target object.

Figure 4 shows the accuracy of boundary-detection models for actions with different durations for several datasets. For all datasets, the accuracy of boundary detection is directly proportional to duration. This suggests that segmentation-based approaches may be fundamentally limited for identification of elemental action sequences at high time resolution. This is very problematic for the proposed dataset StrokeRehab where the action durations are very short (see Figure 3), but it also limits performance on existing benchmarks like 50salads and Breakfast (see Section 3.4). Motivated by this limitation, we propose a sequence-to-sequence method that predicts the sequence of actions directly in the following section.

### 3.4 Proposed methodology: Sequence-to-sequence models

Motivated by the limitations of segmentation-based approaches described in Section 3.3, we propose to directly predict the sequence of actions from the input data using sequence-to-sequence (seq2seq)

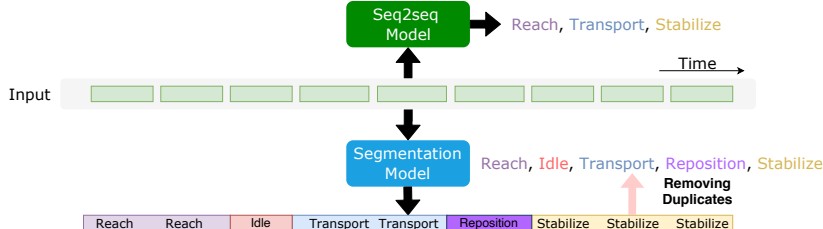

Figure 6: Comparison of sequence-to-sequence (seq2seq) and segmentation models. For an input frame sequence, the segmentation model outputs frame-wise action predictions with the same total length as the input frames. The frame-wise prediction can then be converted to a sequence estimate by removing the duplicates. In contrast, the seq2seq model learns a mapping of a variable-length input sequence to a variable-length output sequence directly.

Table 3: In order to study the effect of distribution shift on StrokeRehab we evaluate a Raw2seq model trained only on healthy subjects (HS), only on stroke patients (SP) and on both (HS+SP) using different test datasets (see Section 2.4). The model trained only on healthy subjects, fails to generalize to the other two cohorts. In contrast, models trained on stroke patients generalize well to healthy subjects. All models have difficulties generalizing to severely impaired patients (but the performance of the HS-trained model is particularly poor) (see Appendix C). We report the mean of the metrics of interest with 95% confidence intervals computed via bootstrapping (see Appendix G.2).

| Tested on | Healthy subjects | Stroke patients | Severely impaired |
|---|---|---|---|
| Trained on | Action Error Rate | Action Error Rate | Action Error Rate |
| Healthy Subjects (HS) | 0.281 (0.263 - 0.299) | 0.405 (0.383 - 0.425) | 0.819 (0.746 - 0.898) |
| Stroke patients (SP) | 0.286 (0.268 - 0.303) | 0.305 (0.284 - 0.324) | 0.612 (0.562 - 0.676) |
| HS + SP | 0.287 (0.269 - 0.304) | 0.297 (0.279 - 0.318) | 0.604 (0.558 - 0.656) |

models inspired by speech recognition (see Figure 6). These models are designed to map input sequences to output sequences of different length, and are therefore well suited to the action-sequence estimation problem. The key idea is to *encode* the input data as a hidden vector of fixed dimension. The hidden vector is then *decoded* sequentially to produce the estimated sequence. During training we apply the seq2seq models on overlapping windows of short duration. During inference, we divide the input data into non-overlapping windows and concatenate the estimates removing duplicates at the boundaries.

Computational constraints limit the window size of inputs to sequence-to-sequence models to 500-1000 time steps. This is sufficient for applications in natural language processing [3] and speech recognition [9]. However, in order to identify high level actions, it may be necessary to model long-term dependencies. To overcome this limitation, we propose a version of our seq2seq model, which uses frame-wise predictions from a segmentation-based model (specifically an MS-TCN model) as inputs. These frame-wise predictions can be interpreted as features capturing long-term dependencies. To differentiate the two versions of our proposed approach we call *Raw2seq* the method that uses raw sequences of sensor data or video features as input, and *Seg2seq* the method that uses frame-wise predictions from a segmentation-based model as input. Appendix G.3 provides a more detailed explanation of our proposed seq2seq models. Additional implementation details are provided in Appendix G.4.

## 4 Experiments and Results

### 4.1 Action sequence identification

We use StrokeRehab to compare the proposed seq2seq methods, described in Section 3.4, to two segmentation-based baselines: MS-TCN [13] and ASRF [24], which was chosen because it out-performs other action segmentation models [57, 38, 17, 10] on existing benchmark datasets (see Appendix G.1). Table 2 reports the results for the *Stroke-patient test set* of StrokeRehab (see Appendix B for additional results on healthy subjects), and also includes results of a comparison based on existing benchmark datasets. To ensure a fair comparison, we optimized the segmentation models using our metric of interest for sequence identification (AER). Interestingly, optimizing this metric,

as opposed to framewise accuracy, substantially boosts the performance of the MS-TCN baseline (see Table 2). Our validation and evaluation procedures are explained in detail in Appendix G.2.

The results in Table 2 indicate that sequence-to-sequence models tend to outperform segmentation-based models. Raw2seq achieves better performance on the sensor data of the StrokeRehab data, where the actions are very localized and do not require modeling long-time dependencies, Seg2seq is superior on the remaining datasets. The baselines achieve edit score values that are close to those of seq2seq on the sensor data of StrokeRehab (but not on the video data), and on 50Salads, but the AER of seq2seq is better. The reason is that, as explained in Section 3.2, the edit score is more lenient with the false positives that tend to be produced by segmentation-based models. Interestingly, the refinement strategies of smoothing and boundary detection (ASRF models) do not improve the baselines on StrokeRehab, highlighting that it contains actions that are qualitatively different from those of the other datasets. Additionally, models selected based on best validation AER generally outperform models selected based on best frame-wise accuracy. This underscores the importance of optimizing metrics that are specifically tailored to sequence identification.

## 4.2 Comparison between data modalities

StrokeRehab contains both video and sensor data, which makes it possible to compare both modalities. Figure 7 shows confusion matrices of the predictions produced by the best models based on video and sensor data respectively. In order to compute AER, there are four components that need to be computed: correctly predicted primitives, substituted primitives, inserted primitives and deleted primitives. The confusion matrices only shows the correctly predicted and substituted primitives. Complete confusion matrices with two more columns namely deleted primitives, and inserted primitives can be seen in Appendix F. The nature of errors of two models were complementary to some extent. For example, the model trained on video data confused reaches with idles, whereas the model trained on sensor data confused reaches with stabilizes. For the video data, confusion occurs mainly between primitives which are usually performed one after the other (eg. idles and reaches). For example, the functional movement that follows idle is a reach 69.3% of the time. This is because the patients tend to be idle right before reaching out to touch an object. In contrast, for the sensor data, there is more confusion between primitives that look similar, such as reaches and transports or idles and stabilizes. Since the nature of errors is disparate, this makes a good case for training multi-modal models that can leverage both modalities to perform the task of sequence estimation.

## 4.3 Studying distribution shift

The different training and test sets in StrokeRehab (see Section 2.4) can be used to evaluate the effect of distribution shift on models trained for action sequence identification. To this end, we trained Raw2seq models on only healthy subjects, only stroke patients, and on both. We then tested these models on the three test sets containing healthy subjects, mildly and moderately impaired stroke patients, and severely impaired patients. Table 3 shows that the model trained only on healthy subjects failed to generalize to the other two cohorts. In contrast, the model trained on stroke patients (mildly and moderately impaired), generalized well to healthy subjects. Therefore, movements observed in stroke patients are relevant to healthy patients, but the opposite is not true. In addition, all models struggled to generalize to the severely impaired patients. This indicates that generalization to different impairment levels is a challenging problem, and that it is paramount to build training sets containing stroke-patient data in order to train models for data-driven rehabilitation. More analysis is provided in Appendix E.

## 4.4 Automatic quantification of rehabilitation dose

In stroke rehabilitation, action identification can be used for quantifying dose by counting functional primitives. Figure 5 shows that the raw2seq version of the seq2seq model produces accurate counts for all activities in the StrokeRehab dataset. Performance is particularly good for structured activities such as moving objects on/off a shelf, in comparison to less structured activities such as brushing, which tend to be more heterogeneous across patients

## 5 Discussion and Conclusion

In this work, we introduce a large-scale, multimodal dataset, StrokeRehab, as a new benchmark for identification of action sequences that includes labled short-duration actions. In addition, we introduce a novel sequence-to-sequence approach, which outperforms existing methods on StrokeRehab as well

(a) Video data                        (b) IMU sensor data

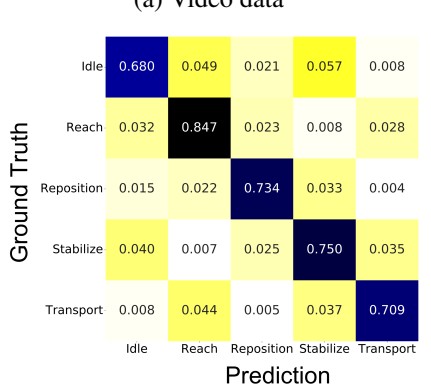 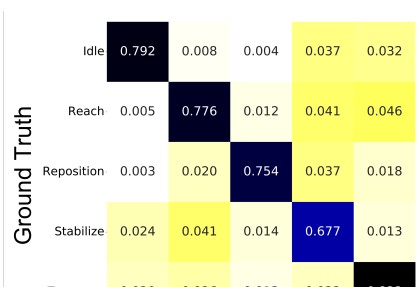

Figure 7: Confusion matrices for the best performing models on the StrokeRehab video and sensor dataset for the stroke-patient test set. The diagonal entries are the fractions of primitives estimated correctly. The off-diagonal entries are the fractions of primitives that were substituted.

as on existing benchmark datasets. A known limitation of sequence-to-sequence approaches is that they have difficulties capturing long-term dependencies. Here, we address this by using the output of a segmentation-based network as an input for the sequence-to-sequence model. An interesting direction for future research is to design sequence-to-sequence models capable of directly learning these dependencies. Our results also show that models based on video and wearable-sensor data have different strengths and weaknesses (see Section 4.4), which suggests that multimodal approaches may have significant potential.

**Acknowledgments**

We would like to thank the volunteers who contributed to label the dataset: Ronak Trivedi, Adisa Velovic, Sanya Rastogi, Candace Cameron, Sirajul Islam, Bria Bartsch, Courtney Nilson, Vivian Zhang, Nicole Rezak, Christopher Yoon, Sindhu Avuthu, and Tiffany Rivera. We thank Dawn Nilsen, and OT EdD for expert advice on the testing battery. This work was supported by an AHA postdoctoral fellowship 19AMTG35210398 (AP), NIH grants R01 LM013316 (AK, KL, HR, CFG, HMS) and K02 NS104207 (HMS), NSF NRT-HDR Award 1922658 (AK, KL, HR, CFG).

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
