# OpenReview forum: "StrokeRehab: A Benchmark Dataset for Sub-second Action Identification"
_NeurIPS.cc/2022/Track/Datasets_and_Benchmarks — NeurIPS 2022 Datasets and Benchmarks _

### Official Review · Reviewer_38gN · 2022-07-26
**This paper proposed a new dataset for sub-second action identification. The dataset consists of a considerable amount of short-duration elemental actions. However, the idea of recognizing actions with high temporal resolution is not novel. Moreover, the writing is not very good, and the experiments are not enough.**

**Rating:** 4
**Confidence:** 4
**Correctness:** The way to construct the dataset soun…

**Strengths:**

1. This dataset consists of a considerable amount of short-duration elemental actions, which required more manual effort to label and more time for training.
2. This dataset contains both video and sensor data, which can leverage both modalities to perform the task of sequence estimation.
3. This dataset provides a benchmark for generalization in the presence of realistic distributional shift and data-driven quantification of rehabilitation dose.


**Weaknesses:**

1. It seems not novel to make a dataset for short-duration action identification since there are many similar practices.
2. There are some writing mistakes, e.g.:
In Table 2, "on both (HS)" should be "on both (HS + SP)".
3. The experiment part is relatively insufficient. The author should supplement the elaboration of the experimental settings, such as the selection of the backbone of the baseline models used, the implementation details of training and testing, etc.
4. Lacking experiments to verify that this dataset can improve the performance of identifying more elemental motions at high temporal resolution. The models trained on this dataset should be tested on the existing datasets and the real scenes. And it would be better to supply the visualized results.
5. The authors spend a little too much space on the action sequence identification algorithm, but the proposed dataset is the theme. To highlight the content of the dataset, the structure of the article should be adjusted appropriately.


**Additional Feedback:**

In Weaknesses, I described in detail what can be improved. The authors can check my comments in Weaknesses for improvement.

**Clarity:**

No, there are some writing mistakes, and the structure of the article should be adjusted appropriately.

**Documentation:**

The dataset are made public. However, some necessary details are insufficient such as information about the data collection and organization.

**Ethics:**

At this moment, I do not find the ethical issue.

**Relation To Prior Work:**

No, some works, e.g. [R1], have the similar practice to this work. Besides, the sub-second actions in the dataset emphasized by the authors seem to have little significance for improving the performance of the model.
[R1] H. Kuehne, A. Arslan, and T. Serre. The language of actions: Recovering the syntax and semantics of goal-directed human activities. In Proceedings of the IEEE Conference on Computer Vision and Pattern Recognition, pages 780–787, 2014.


**Summary And Contributions:**

The authors introduce a large-scale action-recognition dataset called StrokeRehab. The contributions of this paper are summarized as follows:
1. This dataset provides a benchmark for sub-second action identification.
2. The dataset consists of 3,372 trials of rehabilitation activities and 120,891 functional primitives.
3. It provides a benchmark for generalization in the presence of realistic distributional shift.
4. It can be used for quantifying dose by counting functional primitives.

---

> ### Author Response · Authors · 2022-08-26
> **We thank the reviewer for the thoughtful feedback.**
>
> We thank the reviewer for the thoughtful feedback. We have uploaded a revised version where all the changes are highlighted in red. We respond in detail below.
>
> 1. ***It seems not novel to make a dataset for short-duration action identification since there are many similar practices.***
> To best of our knowledge, there are no existing open-source labeled action-recognition datasets with short-duration actions. However, if the reviewer knows of any, we will be very happy to mention them, so please let us know. We have included a detailed comparison to the publicly-available action-recognition datasets, (which we are aware of) that are the most relevant. As seen in Figure 3, these datasets have actions with longer durations. Additionally, the nature of actions is also different. Most of the datasets like Breakfast (R1), 50salads, Jigsaws, FineGYM have labels corresponding to actions like cutting a tomato, zipping up jacket, etc. but our dataset has elemental motions called functional primitives that are of much shorter duration and subtler than functional movements. This is described in detail in Section 2.6 of the main paper.
>
> 2. ***There are some writing mistakes, e.g.: In Table 2, "on both (HS)" should be "on both (HS + SP)".***
> Thank you for pointing this out. We have corrected the typo.
>
> 3. ***The experiment part is relatively insufficient. .... details of training and testing, etc.***
> We have given the details of experiment settings and model architecture in the Appendix E. Specifically, E.1 covers dataset description, E.2 covers the description of validation and evaluation protocols, E.3 describes the seq2seq model architecture and E.4 mentions the implementation details.
>
> 4. ***Lacking experiments to verify that ..... would be better to supply the visualized results.***
> As mentioned in Section 2.6, none of the existing datasets has such short duration granular actions. In fact the nature of the labeled actions is completely different to that of existing datasets. Figure 2 shows a hierarchy of human actions. Existing datasets focus on high level actions associated to particular activities or objects (e.g. cutting vegetables, zipping up jacket), which typically have long durations (seconds or longer) and execute several goals. In contrast, as shown in Figure 3, the actions in the StrokeRehab dataset are much shorter (sub-second). They correspond to functional primitives that execute one goal, and are therefore qualitatively different to the higher-level actions in existing benchmark datasets. Because of this we cannot test the models trained on this dataset on existing datasets. We also would like to emphasize that the data all correspond to “real scenes”: they are acquired from real patients in a real rehabilitation environment.
>
> 5. ***The authors spend a little too much .... structure of the article should be adjusted appropriately.***
> This is a good point. We agree with the reviewer. In order to address this concern we have added more figures and illustrations in the paper to demonstrate different aspects of our dataset. With an increase in the page limit, we have moved a lot of details about sensor and video data collection to the main paper (Section 2.3). We have provided additional details about the sensor data in the supplementary material Appendix G.4, and G.5. We have also added Figure 8 in Appendix G that showcase the placement of nine IMU sensors on the subject’s upper body. In addition, we would like to point out that the novel methodology is included in the paper to showcase how this dataset can motivate methodological advances.
>
> 6. ***No, there are some writing mistakes, and the structure of the article should be adjusted appropriately.***
> We have restructured the paper to include more details of the dataset (in Section 2.3) and which eases the readability of the paper. We have added more figures (Figure 6 to the main paper and Figure 8 to the supplementary material) to improve the illustration and convey the message. Finally, we have added more information regarding the confusion matrices in Appendix E.
>
> 7. ***No, some works, e.g. [R1], have the similar .... pages 780–787, 2014.***
> We do cite the Breakfast dataset released by [R1] (see citation number 32 of the main paper). We also benchmark our proposed approach on the Breakfast dataset and show state-of-the-art performance for action sequence prediction task. In addition, we discuss how the Breakfast dataset is different from ours in Section 2.6, Figure 2, Figure 3 and Figure 4 of the main paper. We are unsure what the reviewer means by “sub-second actions in the dataset emphasized by the authors seem to have little significance for improving the performance of the model”. The goal of this work is to introduce a large-scale, multimodal dataset that can serve as a new action-recognition benchmark that includes elemental short-duration actions labeled at a high temporal resolution, not to improve a specific model.

---

> > ### Author Response · Authors · 2022-08-26
> > **Continue from the previous comment**
> >
> > 8. ***The dataset are made public. ... the data collection and organization.***
> > We have updated the paper with the necessary details. Specifically, the necessary details about the data collection and organization are mentioned in Section 2.3 of the main paper and Appendix G and H of the supplementary material.

---

> > ### Comment · Reviewer_38gN · 2022-08-29
> > **Irrelevant answer. I remain my rating**
> >
> > "To best of our knowledge, there are no existing open-source labeled action-recognition datasets with short-duration actions". You are misunderstanding my concern, I mean what your practical value to construct a dataset for short-duration action identification?
> > The experiments are still insufficient. How about the performance of identifying more elemental motions at high temporal resolution? I want to know “high temporal resolution”, you reply to “none of the existing datasets has such short duration granular actions.”

---

> > > ### Author Response · Authors · 2022-08-29
> > > **Thank you for further clarifying your concerns.**
> > >
> > > Thank you for further clarifying your concerns.
> > >
> > > 1. The practical value for constructing a short-duration action identification dataset is that identifying short-duration actions is of critical importance in stroke rehabilitation. A major clinical question is how many repetitions of functional motions are needed for the recovery of stroke patients. A handful of studies have observed that patients train about 10 times fewer repetitions than what recovering animals receive, suggesting pronounced under-training in human rehabilitation. However, the optimal number of training repetitions to boost recovery remains uncertain in humans. Currently, the best way to quantify rehabilitation is hand tallying. If performed in real-time by the rehabilitation therapist, hand tallying distracts from treatment delivery. If the session is videotaped and annotated offline, tallying is laborious and slow: it takes one hour of manual effort to label one minute of recorded training. Therefore it is crucial to develop methods for automatic identification via machine learning.
> > >
> > > 2. Regarding your second question, by “high temporal resolution” we mean that the dataset requires identifying actions that have durations below one second. Does that answer your question? If the issue is our use of high temporal resolution, we are happy to replace the term with “short duration”.

---

### Official Review · Reviewer_JHMk · 2022-07-26
**Review for StrokeRehab: A Benchmark Dataset for Sub-second Action Identification**

**Rating:** 6
**Confidence:** 4
**Correctness:** I found the claims to be correct

**Strengths:**

1. The motivation of the work is clear. Such a dataset can help the medical community in multiple ways and design better rehabilitation programs in the future.

2. While the absolute number of identities present in the dataset (patients and healthy subjects) is small, I believe the dataset is sufficiently large in the medical domain.

3. The authors manually label the data and cross-check the quality of labels.

4. The authors first show that naively training the current state-of-the-art approaches on this dataset leads to inferior results and proposes a new approach to tackle the task.



**Weaknesses:**

1. I did not find Figure 6 in the paper. I believe it will be the figure which has the architectural definition. The figure is in the supplementary material and should be mentioned properly in the main paper. I believe this is an important figure and thus should be included in the main paper.

2. I believe transformers can also be tried instead of LSTM-based architectures. Data scarcity might be a valid reason not to use transformers which are known to be data hungry, but this should be clearly mentioned and discussed in the paper.

3. While a large amount of sensor data is collected, the information provided in the paper regarding this is less. I believe more information like where the sensors are placed can be provided.









**Additional Feedback:**

None

**Clarity:**

The paper writing can be improved a bit. The figures are too small and often not referred to properly.

**Documentation:**

The dataset collection details

**Ethics:**

The authors followed set ethical protocols. The faces of the subjects were blurred, protecting their privacy.

**Relation To Prior Work:**

The authors explain prior works and datasets properly.

**Summary And Contributions:**

A new dataset called StrokeRehab provides action information for multiple patients rehabilitating from strokes as well as healthy subjects performing similar actions. The dataset contains information from multiple modalities (sensors and video data). The paper also proposes a new algorithm to improve action recognition on this dataset.

---

> ### Author Response · Authors · 2022-08-26
> **We thank the reviewer for the thoughtful feedback.**
>
> Answer: We thank the reviewer for the thoughtful feedback. We have uploaded a revised version where all the changes are highlighted in red. We respond in detail below.
>
> 1. ***I did not find Figure 6 in the paper....thus should be included in the main paper.***
> Thank you for the feedback. We moved the figure to supplementary section due to space constraints. But now that page limit is increased, we have moved the figure (Figure 6) from the supplementary material to the main paper.
>
> 2. ***I believe transformers can also .... and discussed in the paper.***
> We did perform some preliminary experiments with transformers but found that they do not perform as well as LSTM models. However, we agree that this is a promising research direction, which will be supported by the public release of the dataset.
>
> 3. ***While a large amount of sensor .... placed can be provided.***
> With an increase in the page limit, we have moved a lot of details about sensor and video data collection to the main paper (Section 2.3). We have provided additional details about the sensor data in the supplementary material Appendix G.4, and G.5. We have also added Figure 8 in Appendix G that showcase the placement of nine IMU sensors on the subject’s upper body.
>
> 4. ***The paper writing can be improved a bit. The figures are too small and often not referred to properly.***
> Thank you for the feedback. We have increased the figure size to make it more presentable and corrected all incorrectly referenced figures/tables. We have also reorganized the contents of the paper to ease the readability of the paper.

---

### Official Review · Reviewer_CP54 · 2022-07-27
**Detailed annotations for sub-second action sequence recognition**

**Rating:** 7
**Confidence:** 3
**Clarity:** Yes.

**Strengths:**

The paper contextualizes its results well in the field and is well described.

**Weaknesses:**

While I believe the dataset is largely sound, I have multiple comments that I would like to see clarified and resolved.

### Data Collection

•	Were the video and sensor data synchronized?

•	Can you comment on how similar the video and sensor data were from subject to subject and day to day. Knowledge of domain differences across video recordings (e.g. shifts of perspective) will affect their generalizability. Raw IMU data will depend on the subject size.

### Rehabilitation Applications

•	I was confused about how AER was chosen as the metric of choice, as opposed to TPR/FDR of actions, or more general task-based quantities (eg effector velocity). The description is cursory, and it is important to communicate for non-domain experts. It also affects the conclusions, as segmentation models were more performant on the basis of TPR.

•	Is the accuracy of these models sufficient to reach desired capacity for physical therapy applications? Much of the scope of this application track is so domain-agnostic experts , and it is not clear what AER is satisfactory for applications.

•	Related to the above: what is the inter-human reliability in the AER?

•	L295 “detailed count-based results are provided in Appendix D” I am not sure what this refers to.

### Domain shift:

* Why is there asymmetry in the transfer of models in Table 1? Why does stroke data generalize to health but not vice versa? One trivial reason is that because there are more users in the stroke dataset, they may ‘cover’ domains better, e.g. contain participants of a greater range of sizes and shapes and video conditions, improving the machine learning transfer. Were the comparisons in Table 1 balanced?

* Can you comment on how labels were made across domains, that is, were different definitions based e.g. on kinematics made for labeling stroke primitives given that they may be overtly different kinematically

* What is the quality of action recognition models on examples from this dataset, e.g. given individual pre-segmented videos? Is the issue with the seq-seq models and segmentation models that actions are hard to recognize or that they are hard to segment?

* Table 3: Why don’t confusion matricies sum up to one row-wise or columnwise to 1? Also it should be clarified whether the matricies are normalized row-wise or column-wise.



**Additional Feedback:**

None.

**Correctness:**

I think the dataset methodology and benchmarking is largely sound, although see specific points above.

**Documentation:**

Yes.

**Ethics:**

No concerns.

**Relation To Prior Work:**

Yes.

**Summary And Contributions:**

This paper describes the StrokeRehab Dataset for action sequence recognition. The dataset consists of sub-second 3,372 trials (120,891 functional ‘primitives’) in 71 subjects. Data was recorded with 9 IMUs on the upper extremities, and featurized using the IMU acceleration, quaternion output, and joint angle representation from a proprietary algorithm. Video data was also recorded from two views as well. The data was labeled in a highly granular manner, under supervision of a domain-expert. Using this dataset, the authors tested different approaches for action sequence recognition, finding sequence-to-sequence models outperformed segmentation based models for this data as well as others in the field.

Overall I was impressed by the granularity of action labeling in the paper and the novelty of the application to the community. I think the authors make interesting and likely valid conclusions about the utility of action segmentation vs. seq-2-seq approaches for action labeling for short primitives, which speaks to the potential for the reuse of the dataset in the community. So I am open to accept this manuscript, however I have reservations below that may affect this decision. While I was impressed by the annotation density, I do have multiple questions that I would like to be resolved before acceptance.

---

> ### Author Response · Authors · 2022-08-26
> **We thank the reviewer for the thoughtful feedback**
>
> Answer: We thank the reviewer for the thoughtful feedback. We have uploaded a revised version where all the changes are highlighted in red. We respond in detail below.
>
> 1. ***Were the video and sensor data synchronized?***   Yes. The video and sensor data is generally recorded at 100 Hz, except for some recordings where the video is recorded at 60 Hz. In that case, the videos can still be matched to the sensor data by sub-sampling the sensor data.
>
> 2. ***Can you comment on how similar ... will depend on the subject size.***   All the videos were recorded from two perspectives (camera), one camera looking down on the subject giving a top-down view and another camera looking towards the subject giving a head-on front view. All the videos are recorded in a rehabilitation gym setting. Therefore, the lightning and background are controlled and do not exhibit much variation.
>
>
> Regarding the variability of the IMU data with size, body shape does not produce substantial variability in the data due to the postprocessing: Angular velocities are converted to sensor-centric unit quaternions, representing the rotation of each sensor on its own axes, with coordinate transformation matrices. In addition, proprietary software (Myomotion, Noraxon) generates 22 anatomical angle values using a rigid-body skeletal model scaled to patient height (hence making the data invariance to patient height). We have updated the main paper with the above information in Section 2.3 (Description of kinematic data)
>
> 3. ***I was confused about how AER was chosen ... performant on the basis of TPR.***
> TPR and FDR are very useful metrics, but they need to be evaluated in combination, which makes it difficult to perform direct model comparison. The segmentation-based models indeed have slightly higher TPR than sequence-to-sequence models (0.79 vs 0.767), but their main weakness is their higher FDR (0.201 vs 0.166), which is a result of the over-segmented noisy outputs produced by these models (see Section 3.3). We have found that AER is a reasonable single metric for overall performance, but we definitely agree that TPR and FDR are very useful and should be reported. We agree that this requires additional clarification and have added a comment about it in Section D.
>
> 4. ***Is the accuracy of these models ... clear what AER is satisfactory for applications.***
> That is very good question. In Section 4.4, we discuss how accurate the model is for performing clinically-meaningful quantification of rehabilitation dose. Figure 5 shows that the seq2seq model produces very accurate counts for activities such as moving objects on/off a shelf (0.09 MAPE), moving object on a table (RTT 0.02 MAPE), feeding (0.02 MAPE) and drinking (0.06 MAPE). However there are some activities like face wash, and deodorant where the MAPE is not as low. We hope that the release of the dataset will contribute to the development of methods that will further increase performance for these activities.
>
> 5. ***Related to the above: what is the inter-human reliability in the AER?***
> We have computed the inter-human reliability in terms of AER. It equals 0.021. We have added this information in Section 2.5 of the paper.
>
> 6. ***L295 “detailed count-based results are provided in Appendix D” I am not sure what this refers to.***
> Thank you for pointing it out. It is typo. We decided to convert the table to a figure (Figure 5 of the main paper) and remove the table from the Appendix. But we forgot to remove the sentence. It has been removed.
>
> 7. ***Why is there asymmetry in the ....Were the comparisons in Table 1 balanced?***
> This is an interesting question, which illustrates the importance of creating datasets that allow to evaluate the type of domain shift present in rehabilitation data. As explained above, there is not substantial variability in the video and sensor data. The asymmetry is due to the fact that healthy subjects perform much more regular movements than stroke patients, which are much more heterogeneous (there are many fewer ways to perform a functional movement well, than to perform it wrong). The stroke dataset contains many examples of “normal” movements (especially from the mildly impaired patients), which explains generalizability to healthy patients. However, the healthy patients' data lack examples of the abnormal movements that are needed to perform well on stroke patients.
>
> 8. ***Can you comment on how labels .... overtly different kinematically***
> The labels were generated by viewing the video at 0.1x and labeling the start and end of the primitive. As the IMU data was synchronously recorded, it was also labeled simultaneously. Therefore, the label definition for both modalities is the same.

---

> > ### Author Response · Authors · 2022-08-26
> > **Continue from the previous comment**
> >
> > 9. ***What is the quality of action ....recognize or that they are hard to segment?***
> > This is a very interesting question. We believe that the main difficulty is segmentation, especially for very short duration actions. Figure 4 shows that as the action duration shortens, the accuracy of detecting the boundaries is drastically reduced. That said, we think the experiment suggested by the reviewer would be insightful and we will include the results in the appendix.
> >
> > 10. ***Table 3: Why don’t confusion matrices ....column-wise.***
> > In order to compute AER, there are four components that need to be computed: correctly predicted primitives, substituted primitives, inserted primitives and deleted primitives. The confusion matrix only shows the correctly predicted and substituted primitives. There are two more columns namely deleted primitives, and inserted primitives. When we add correctly predicted, substituted and inserted primitives adds up to 1 (i.e. total number of GT primitives).  The complete matrices for IMU and video data have been added to Appendix E (Tables 9 and 10). Since we were discussing the potential confusion between two primitives and to keep the visual simple, we decided to only showcase the correctly predicted and substituted primitives.

---

> > > ### Comment · Reviewer_CP54 · 2022-08-28
> > > **Thank you for the reply**
> > >
> > > Thanks for the reply. Overall the clarifications help and I appreciate highlighting the changes to the manuscript in red. Overall I think the manuscript is worthy of inclusion and I'll leave my score unchanged, but I would appreciate close attention to all of the reviewer's comments. In general I agree with the sentiment that the dataset should be more the focus than the algorithm, and that the manuscript itself is a bit unpolished and takes a bit of effort to follow, with lots of digging in the Appendix. I'll respond to more specific points here.
> > >
> > > 1. The manuscript now lists this at 100 Hz. Please make sure that the correct framerate(s) are stated and that common timestamps are provided for potential users of the dataset.
> > >
> > > 10. The definition of what is shown as a confusion matrix needs to be made clear in the text and not just the appendix.
> > >
> > > 5. Include estimates of standard deviation or other measures of variance for this metric and others throughout the paper.

---

> > > > ### Author Response · Authors · 2022-08-29
> > > > **Thank you for the encouraging review and feedback.**
> > > >
> > > > Thank you very much for the feedback. We will further polish the manuscript and address these three points, which we agree with.

---

### Official Review · Reviewer_cXwr · 2022-07-27
**Micro-action dataset for activity of daily living rehabilitation**

**Rating:** 6
**Confidence:** 3

**Strengths:**

* The dataset proposed contains data from a large number of patients.
The dataset has also been annotated by several people, following a precise protocol, with a good interannotator agreement score.

* The algorithms proposed are contrasted against state of the art algorithms on both the proposed dataset and other datasets

**Weaknesses:**

* the annotators training and annotation protocol is very precise but I have doubts about its adequacy, the process needs to be justified. The training seems to enable all annotators to acquire the same definition of the actions and the labeling process. However, I have concerns this lengthy process also erases the inter-annotator variability : the "coders" could end up emulating the expert's annotation instead of labeling by his own judgement, and this would erase inter-annotator variability. The labeling by experts would be more interesting than by coders, even with a smaller number of labelers.

* the approaches proposed lack a detailed description in the article. A high-level explanation and motivation are given, but the final algorithms themselves are not described in the article, only in the supplementary material.

* the dataset aims at segmenting elementary actions that are quite similar. However, the data acquisition uses wearable sensors that are placed on the hands : they and the straps can be bulky and change the natural motion of the subjects. This change can have an impact especially on similar movements. Has this impact been evaluated ?







**Additional Feedback:**

none

**Clarity:**

* I did not find in the article the number of annotators.
* In section 2.4, what is the definition of "severely impaired" subjects ? How is the severity assessed, and which is the threshold ?
* what does “removing duplicates at the boundaries“ mean ?
* the word "action" is used in this article with several meanings: from movements or simple actions (reaching, transporting) to activities or complex actions (brushing,feeding). But the wording is not always clear which "action" it is currently about.


**Correctness:**

* In table 2, the model trained on SP has the following results : the accuracy of the test on healthy subjects is better than tests on stroke patients. This seems strange that the model performs worse on the distribution it was trained on. This should be addressed, and does not show the distributional shift, as stated in the article
Moreover, the model trained on HS+SP has better results than the model trained only on SP when tested only on SP. Again, this can seem puzzling and should be addressed, as this is not a sign of the distributional shift described in the article.

* In section 4.2, l. 271:"For the video data, confusion occurs mainly between primitives which are usually performed one after the other "  :it does not sound a convincing hypothesis, it seems based on only 1 example. What is the usual order of actions ?

* In table 3, the confusion matrices do not seem to be normalised, neither by row nor by column.

* there are a few errors in the references, for instance "Table 4.3"

**Documentation:**

* I did not see the licence for this dataset, only the licences for the datasets this paper compares to.


**Relation To Prior Work:**

* The authors argue that existing datasets do not label short-duration actions. However, the number of datasets of actions using IMU and/or videos is very high. Therefore, a more precise definition of the subfield to compare to, would have been necessary. Currently, the listed datasets are very few compared to the state of the art. Moreover, I feel that a focus could have been developed on medical datasets for physical rehabilitation.


**Summary And Contributions:**

This article proposes a dataset for identifying short duration actions for physical rehabilitation that contains both video and IMU data. THe dataset includes data from stroke-impaired patients and healthy subjects performing activities of daily living, and labels at a high temporal resolution.

Along with the dataset, the article proposes a novel sequence-to-sequence approach to identify actions, and presents the perfomance on this dataset and other datasets of two versions of algorithm from their proposed approach and compared with state-of-the art algorithms.

---

> ### Author Response · Authors · 2022-08-26
> **We thank the reviewer for the thoughtful feedback.**
>
> We thank the reviewer for the thoughtful feedback. We have uploaded a revised version where all the changes are highlighted in red. We respond in detail below.
>
> 1. ***The adequacy of the annotators***   The rationale for training annotators was to bring them up to the level of an expert, which we confirm with their high annotator-expert reliability. Emulating an expert’s annotation is in fact the goal, and was necessary to increase throughput of data labeling. To this end, we provided rigorous training to the coders and we also asked the experts to check the one-third of the labels annotated by the coders. The ground truth labels can thus be considered “expert labels”.
>
> 2. ***The approaches proposed lack a detailed description in the article.*** We have prioritized describing the dataset in the main paper, since this is the main focus of the paper. However, we agree with the reviewer that it would be helpful to include some more details in the main paper (to the extent that is is possible given the space constraints), so we have included a detailed diagram explaining the methodology (Figure 6). As pointed out by the reviewer, the approaches are described in detail in the Appendix E.
>
> 3. ***This change can have an impact especially on similar movements. Has this impact been evaluated?***  This is an important point, we thank the reviewer for drawing our attention to it. The sensors are lightweight (34 g) and small (length: 37.6 mm; width: 52.0 mm; height: 18.1 mm). They are adhered to the back of the hand with thin tape that does not interfere with finger movement or grasp. Similarly, the straps holding the sensors to the forearms and arms do not cross any joint. Therefore neither the location nor the methods used to affix the sensors interfere with natural motion. We have added this important information to Section 2.3 of the main paper.
>
> 4. ***Question on the results for model trained on SP and SP+HS***  This is indeed an interesting observation, and illustrates the importance of creating datasets that allow to evaluate the influence of domain shift on action recognition models. The general structure of the functional movements is similar in healthy subjects and stroke patients, although it tends to be more regular in healthy patients. Therefore the model trained on stroke patients is able to learn “normal” functional movements that enables it to perform well on healthy subjects. The reason why it does not perform as well on the test stroke patients is that the movements of stroke patients tend to be very heterogeneous, which makes action recognition more challenging than for healthy subjects (especially for the test patients with higher impairment). We observe the same phenomenon for the segmentation-based models (see Table 5).
>
> 5. ***Confusion occurs mainly between .. does not sound a convincing hypothesis, it seems based on only 1 example. What is the usual order of actions?***  Thank you for raising this point. We were not sufficiently clear and have edited to paper to address this in Section 4.2. The functional movement that follows idle is a reach 69.3% of the time. This is because the patients tend to be idle right before reaching out to touch an object. Similarly the stabilize functional movement assumes that an object has been grasped, so it is typically preceded by transport.
>
> 6. ***Confusion matrices do not seem to be normalised, neither by row nor by column.***  This is a good point. In order to compute AER, there are four components that need to be computed: correctly predicted primitives, substituted primitives, inserted primitives and deleted primitives. The confusion matrix only shows the correctly predicted and substituted primitives. There are two more columns namely deleted primitives, and inserted primitives. When we add correctly predicted, substituted and inserted primitives, they add up to 1 (i.e. total number of GT primitives).  The complete matrices for IMU and video data have been added to the Appendix E (Table 9 and 10). Since we were discussing the potential confusion between two primitives and to keep the visualization simple, we decided to only show the correctly predicted and substituted primitives.
>
> 7. ***The number of annotators.*** There were two experts and seven annotators. We have added this information to Section 2.5.
>
> 8. ***The definition of "severely impaired" subjects.***  As explained in Section 2.4, "severely impaired" subjects are defined as the subjects with impairment FM scores [1]  in the range 8-23.
>
>
> [1] Fugl-Meyer, Axel R., et al. "A method for evaluation of physical performance." Scand J Rehabil Med 7.1 (1975): 13-31.
>
> 9. ***What does “removing duplicates at the boundaries“ mean?*** The data are separated into frame windows, so the last action detected in a frame is the same as the first action in the next frame. We have added a diagram in Figure 6 to make this clearer.

---

> > ### Author Response · Authors · 2022-08-26
> > **Continue from previous comment**
> >
> > 10. ***The word "action" is used in this article with several meanings*** Thanks for pointing this out, we agree it would be more consistent to always denote ‘simple actions (reaching, transporting)’ as ‘primitives’ while the complex actions (brushing, feeding) as high-level actions.
> >
> > 11. ***The license for this dataset***  We have added the license to the supplementary material Appendix H in the updated version. It is also available on https://simtk.org/projects/primseq after making an account with SimTK.
> >
> > 12. ***The authors argue that existing datasets ... been developed on medical datasets for physical rehabilitation.***
> > We have included a detailed comparison to the publicly-available action-recognition datasets, (which we are aware of) that are the most relevant. If the reviewer knows of additional publicly-available labeled action-recognition datasets with short-duration actions, we will be very happy to mention them. As far as we know, this is the first labeled short-duration publicly-available dataset of sensor and video data for physical rehabilitation, but if the reviewer knows of any other we will also be very happy to include them.

---

### Official Review · Reviewer_WdkF · 2022-07-27
**StrokeRehab dataset and sequence-to-sequence approach for sub-second action identification**

**Rating:** 6
**Confidence:** 3

**Strengths:**

The proposed dataset is interesting and seems challenging compared to existing ones, in particular for the task of sub-second action identification. As it is multimodal, I think it could be beneficial for the research community in both the field of computer vision and time series analysis. In addition, the datasets also propose additional challenges like distributional shift and quantification of rehabilitation dose that are very interesting.
Moreover, the proposed approach for high resolution action identification sounds good and promising.


**Weaknesses:**

The fact that authors simultaneously propose a new dataset and a novel approach can also be seen as a weakness. Indeed, it results in partial information for both parts. While the supplementary material is significant and provides useful information, the necessity to make back-and-forth between the paper and the appendix does not facilitate the reading and the whole understanding. It would maybe be a good idea to focus the paper on the dataset (with a benchmark using existing methods) as it is the scope of this track, and let the novel approach for a further paper.

**Additional Feedback:**

Please find below some additional minor remarks/suggestions:
- I suggest authors carefully read the paper to correct some remaining typos
- First sentence of the abstract and the introduction are the exact same
- I think subsection 2.3 can be merged with subsection 2.1
- Table 3 should be Figure 3 as it shows confusion matrices


**Clarity:**

The paper is clear but could be better organized. Moreover some typos and grammatical errors remain.

**Correctness:**

For the dataset part, it seems that it has been constructed in a sound way.
For the benchmark part, I think the robustness could be improved regarding the train/test split. Indeed, If I understood well, either for healthy or stroke patients, a single train/test split has been considered. As a result, the compared approaches are evaluated on this particular split. Hence, it is difficult to conclude about the reported performances, if they are general or valid only for this particular case. I think considering different splits in a cross-validation scheme would add significant robustness. A common scheme in similar cases is the use of the one-subject-out cross validation.

**Documentation:**

For the dataset part, sufficient information is provided through the paper and the appendix. Even if the dataset is not completely publicly available (it will be the case if the paper is accepted), the provided samples allow us to understand the data.
For the benchmark, it seems that the code and the implementation details provided in the appendix are suitable for reproducibility.

**Ethics:**

I do not observe any ethical concerns.

**Relation To Prior Work:**

The differences between the proposal and previous contributions are mentioned. However, most of these discussions are included in the appendix and it does not make the reading smooth.

**Summary And Contributions:**

In this paper, authors introduce a new multimodal dataset called StrokeRehab. It includes inertial measurements and video data of several daily activities performed by 51 stroke impared patients and 20 healthy subjects. Sequences are labeled at a high temporal resolution for annotating elemental short-duration actions. In addition, a novel approach dedicated to high resolution action identification is proposed. Experiments on the proposed dataset as well as existing ones suggest that the proposed approach is suitable for the task in comparison to state-of-the-art.

---

> ### Author Response · Authors · 2022-08-26
> **We thank the reviewer for the thoughtful feedback**
>
> We thank the reviewer for the thoughtful feedback. It will help us improve the manuscript. We have uploaded a revised version where all the changes are highlighted in red. We respond in detail below.
>
> 1. ***Focus the paper on the dataset …novel approach for a further paper*** We completely agree with the reviewer and have reorganized the content to address this. We have moved many of the details about the sensor and video data collection to the main paper (Section 2.3). We have provided additional details about the sensor data in the supplementary material Appendix G.4, and G.5. We have also added Figure 8 in Appendix G that showcase the placement of nine IMU sensors on the subject’s upper body. In addition, we would like to point out that the novel methodology is included in the paper to showcase how this dataset can motivate methodological advances.
>
> 2. ***The robustness could be improved regarding the train/test split*** This is a great point. The test set is carefully constructed to ensure that it is representative of the patient population (e.g. in terms of impairment level and paretic side). We believe that having a completely held-out representative test set is important for rigorous comparison. We have described the process of creating the test set in Section 2.4. That said, we completely agree with the reviewer that it is important to evaluate model robustness to different splits. When training the models we performed 4 different training/validation splits. In order to show robustness, we will report the validation results on all splits in the appendix.
>
> 3. ***The differences between the proposal ... and it does not make the reading smooth***
> As explained above, we have reorganized the paper to provide a more coherent description of the dataset in the main paper. Section 2.6 includes a detailed comparison to previous datasets. The beginning of Section 3.3 has a description of the key limitations of the current state of the art, which provide the necessary context for the main paper,  and refer to Appendix A for a more comprehensive description, which is included for completeness but could not fit in the main paper due to space constraints.
>
> 4. ***Additional feedback on typos and content*** We thank the reviewer for pointing out these out, we have addressed them.

---

### Official Review · Reviewer_zL1Z · 2022-07-28
**Short-duration action dataset collected by cameras and IMUs**

**Rating:** 7
**Confidence:** 3
**Correctness:** Yes, the claims in the paper are corr…
**Clarity:** The paper is well written.

**Strengths:**

This dataset is different from other existing datasets and focuses on actions in short duration. This new way of action labeling provide a different view of behaviors that can be helpful for rehabilitation after stroke. With this large-scale dataset, a new action-recognition benchmark is presented using several previous segmentation-based approaches and newly developed sequence-to-sequence models inspired by speech recognition. The result also shows that the model trained on impaired patients can also be applied to healthy subjects, but the opposite is not working. And this action recognition is shown to be useful for quantifying does by counting these short-duration actions (functional primitives).

**Weaknesses:**

The paper presents a dataset including video data and motion data from IMUs. However, the motivation is not clear. What is the advantage of adding IMU sensor data? Setting up two cameras is very easy, but attaching IMUs to a human body is not very easy. It is also useful to show whether adding multiple IMUs can affect action execution. From Figure 1, it is clear that tow IMUs are attached to human hands and this setup may have some effect on impaired patients. The paper states that it introduces a multimodal dataset but the benchmark is done respectively and the paper doesn't show the advantage of the multimodal data.

**Additional Feedback:**

The website of the dataset can be improved. Adding some data visualization and videos are helpful.

**Documentation:**

Yes, it has good documentation.

**Ethics:**

I don't see major ethical concerns.

**Relation To Prior Work:**

Yes, this paper clearly discussed the contribution compared with previous works.

**Summary And Contributions:**

This dataset includes daily living action data collected by two cameras and several IMUs from 51 stroke-impaired patients and 20 healthy subjects. Different from previous human action datasets, this dataset focuses on short-duration action (functional primitives) recognition, including reach, reposition, transport, stabilization, and idle. The labeling requires a significant amount of effort from trained annotators. The motivation is to facilitate the rehabilitation process after stroke that requires repetition of sub-second actions. In addition, a new action sequence identification method is presented that can outperform other approaches and the method is validated on the presented dataset and other previous video datasets.

---

> ### Author Response · Authors · 2022-08-26
> **Thank you for the encouraging feedback**
>
> We thank the reviewer for their encouraging feedback. We believe it will improve the manuscript. We have uploaded a revised version where all the changes are highlighted in red. We respond in detail below.
>
> 1. ***What is the advantage of adding IMU sensor data***
> In realistic rehabilitation settings videos have limited applicability due to occlusions, which necessitate multiple viewpoints. Wearable IMUs are small (length: 37.6 mm; width: 52.0 mm; height: 18.1 mm) and lightweight (34 g) enough to enable the patients to perform rehabilitation, so they are a promising option for data acquisition. An important aspect of our dataset is that it enables comparison between these different data modalities, which could lead to useful conclusions for clinical applications.
>
> 2. ***Adding multiple IMUs can affect action execution***   The sensors are lightweight (34 g) and small (length: 37.6 mm; width: 52.0 mm; height: 18.1 mm). They are adhered to the back of the hand with thin tape that does not interfere with finger movement or grasp. Similarly, the straps holding the sensors to the forearms and arms do cross joint. Neither location nor the methods used to affix the sensors are expected to interfere with natural motion. We have added this information to Section 2.3 of the main paper.
>
> 3. ***Advantages of multimodal dataset*** As explained above, a crucial open question for data-driven rehabilitation is what data modality to use. Therefore it is important to create open datasets that enable comparisons across modalities. Our results show that wearable sensors and video are both viable data-acquisition strategies, and that their errors are complementary to some extent (see Section 4.2).
>
> 4. ***The website of the dataset can be improved*** We thank the reviewer for the suggestion and will improve the website by adding visual examples of our data and annotating these examples with model predictions.

---

### Meta-Review · Area_Chair_JmPR · 2022-09-09

**Recommendation:** Accept
**Confidence:** 4

**Metareview:**

This submission receives reviews from 6 different reviews. Most reviewers (5/6) appreciate the contribution of the new dataset. They acknowledge that the problem setup is interesting and the dataset may be useful for different research communities: computer vision, time series analysis, medical. On the other hand, reviewer 38gN concerns about the novelty of the proposed dataset. AC reads all reviews and comments, and discussions, and is convinced that the proposed dataset will provide a useful benchmark for research, thus AC recommends to accept this submission as a poster. AC recommends the authors to incorporate all suggestions from reviewers for the final camera version.

---

### Decision · Program_Chairs · 2022-09-16

Accept